# C∙∙∙O and Si∙∙∙O Tetrel Bonds: Substituent Effects and Transfer of the SiF_3_ Group

**DOI:** 10.3390/ijms241511884

**Published:** 2023-07-25

**Authors:** Zhihao Niu, Qiaozhuo Wu, Qingzhong Li, Steve Scheiner

**Affiliations:** 1The Laboratory of Theoretical and Computational Chemistry, School of Chemistry and Chemical Engineering, Yantai University, Yantai 264005, China; nzh19981019@s.ytu.edu.cn (Z.N.); wqz1143280082@163.com (Q.W.); 2Department of Chemistry and Biochemistry, Utah State University, Logan, UT 84322, USA

**Keywords:** noncovalent bonds, AIM, energy decomposition, charge transfer

## Abstract

The tetrel bond (TB) between 1,2-benzisothiazol-3-one-2-TF_3_-1,1-dioxide (T = C, Si) and the O atom of pyridine-1-oxide (PO) and its derivatives (PO-X, X = H, NO_2_, CN, F, CH_3_, OH, OCH_3_, NH_2_, and Li) is examined by quantum chemical means. The Si∙∙∙O TB is quite strong, with interaction energies approaching a maximum of nearly 70 kcal/mol, while the C∙∙∙O TB is an order of magnitude weaker, with interaction energies between 2.0 and 2.6 kcal/mol. An electron-withdrawing substituent on the Lewis base weakens this TB, while an electron-donating group has the opposite effect. The SiF_3_ group transfers roughly halfway between the N of the acid and the O of the base without the aid of cooperative effects from a third entity.

## 1. Introduction

The tetrel bond [1] (TB) is defined as an attractive interaction between the tetrel center (group 14 element) of an electron-deficient Lewis acid and an electron-rich Lewis base. TBs have an unparalleled importance in molecular recognition [2] and crystalline materials [3] since they can be used as a new molecular linker in crystal engineering. Therefore, the structure, strength, properties, and nature of TBs have been extensively studied [4,5,6,7,8,9,10]. When a tetrel atom is adjacent to an electron-withdrawing atom or group, a region with a positive electrostatic potential is found on its surface. When the tetrel atom is sp^3^- and sp^2^-hybridized, such regions are called σ-holes and π-holes, respectively [11,12]. Furthermore, sp^3^-hybridized tetrels usually form weaker tetrel bonds than do sp^2^-hybridized tetrels. In most cases, the tetrel bond magnifies in the order C << Si < Ge < Sn < Pb due to the fact that the higher the atomic number, the smaller the electronegativity, and the greater the polarization [11]. The carbon atom has difficulty in forming a TB unless it is attached to a strong electron-withdrawing group, or the Lewis base with which it interacts is particularly strong. For example, the methyl carbon atoms of CH_3_OH and CH_3_F can form weak tetrel bonds with some Lewis bases such as H_2_O [13]. Although the tetrel bond involving the carbon donor is relatively weak, it plays an important role in protein–ligand systems [14,15] and chemical reactions [16].

Due to its greater electronegativity and lower polarizability, the carbon atom is rarely involved in strong TBs, which are also known as carbon bonds, if the carbon atom acts as a Lewis acid. Even so, carbon bonding has been observed in different systems. Weak Ar∙∙∙C interactions were first detected in Ar∙∙∙propynyl alcohol complexes by microwave spectroscopy and ab initio calculations [17]. Mani and Arunan carried out a theoretical study of carbon bonding in complexes of methanol and methyl fluoride, where the methyl group is attached to an electron-withdrawing group and placed in conjunction with a neutral base, finding interaction energies of ~2 kcal/mol [13]. The prevalence of such carbon bonding was further demonstrated in the solid state using the Cambridge Structural Database (CSD) and charge density analysis [18]. Similarly, the -CF_3_ group in ArCF_3_ is able to participate in carbon bonding if the electron donor atom is not combined with active hydrogen atoms [19]. Otherwise, it favors the formation of weak hydrogen bonds through the F atoms [20].

The CF_3_ group is common in drug molecules and it is involved in the recognition of the activity of medicament [21,22]. Thus, researchers have designed numerous trifluoromethylating reagents to introduce a CF_3_ group into a target molecule [23,24,25]. It has been demonstrated that this group plays a dual function in noncovalent interactions since its C atom provides a σ-hole to join a tetrel bond and its F atoms are functioning electron donors [26]. There have been attempts to limit the group’s function to only the TB. One approach is to select an electron donor such as NCLi without active hydrogen atoms to avoid the formation of a H-bond to F [4]. An alternate strategy involves deepening the σ-hole on the C atom, which can be realized by adding a positive charge to the tetrel donor molecule or adjoining an electron-withdrawing group to the C atom [13,20]. In tetrazole-CF_3_∙∙∙NCLi, where both methods were employed, a pure tetrel bond was achieved with an interaction energy of 3.79 kcal/mol [27].

In the protodesilylation reaction of phenyltrifluorosilane (PTS) with 8-hydroxyquinoline, a C–Si bond cleavage was found and the –SiF_3_ group migrated from PTS to 8-hydroxyquinoline, resulting in a pentacoordinated silicon complex [28]. Such pentacoordinated silicon complexes have been synthesized and characterized [29,30,31,32]. They were thought to be an intermediate along the S_N_2 reaction pathway and the authors pointed out that the nature of the leaving group is a key factor in determining the degree of Si–O bond formation [33]. Past work has considered the transfer of an sp^3^-hybridized silicon group between two atoms that are facilitated by cooperativity with another interaction, such as the cation–π interaction, triel bond, and beryllium bond [27,34,35]. A major question emerges as to whether such a migration can occur without benefit of cooperativity engendered by a third unit.

In order to address this question, the present work begins with the construction of what may be a powerful tetrel-bonding C or Si atom. The NH hydrogen of the saccharin molecule is replaced by both a CF_3_ and a SiF_3_ group to compare C and Si in this regard. This molecule, designated here as S-TF_3_, contains the elements of a strong TB, in that the N of the TF_3_ group is attached to electron-withdrawing CO and SO_2_ groups (see Figure 1). As a Lewis base partner, pyridine-1-oxide (PO) contains an O atom with a partial negative charge. The ability of this molecule to participate in a TB is modulated by placing various substituents at the position para to N-O. The list of substituents CH_3_, OH, OCH_3_, NH_2_, Li, NO_2_, CN, and F is of varying degrees of both electron-withdrawing and -donating potential so as to cover a wide gamut of base strength. Questions of interest first concern how the nature of each substituent affects both the C∙∙∙O and Si∙∙∙O tetrel bonds. How strong are these TBs and are they constituted of the same components as are generic noncovalent bonds? Another primary issue involves the possibility that either the CF_3_ or SiF_3_ group might transfer toward the O atom of the base, and by how much. Can such a transfer occur even in the absence of a third molecule and its cooperative influence? 

## 2. Theoretical Methods

The geometries of both monomers and complexes were optimized at the M06-2X/aug-cc-pVDZ level. This method has been used to study various tetrel-bonded complexes due to its accuracy and ability to incorporate dispersive effects [10,36,37]. All geometries corresponded to true minima on the potential energy surfaces since no imaginary frequencies were found when performing harmonic frequency calculations at the same level. The interaction energy (E_int_) was calculated as the energy difference between the complex and the monomers frozen in their geometry within the complex. When the optimized geometries of monomers are utilized, this quantity equates to the binding energy (E_b_). Both E_int_ and E_b_ were corrected for basis set superposition error (BSSE) using the method proposed by Boys and Bernardi [38]. All calculations were performed using the Gaussian 09 program [39].

The molecular electrostatic potential (MEP) for each monomer on a 0.001 a.u. isodensity surface was calculated at the M06-2X/aug-cc-pVDZ level using the wave function analysis surface analysis software (WFA-SAS) [40]. The electron density (ρ), Laplacian (∇^2^ρ), and total energy density (H) at each bond critical point (BCP) were evaluated by the Multiwfn [41] procedure using Bader’s atom-in-molecule (AIM) theory [42]. Charge transfer and orbital interactions between the electron donor and acceptor in the complexes were analyzed using the natural bond orbital (NBO) method [43]. The role of orbitals in the complexes was analyzed by the extended transition state–natural orbitals for chemical valence (ETS-NOCV) [44]. The interaction energy was decomposed into five physically significant components at the M06-2X/aug-cc-pVDZ level using the Local Molecular Orbital Energy Decomposition Analysis (LMOEDA) method [45] in the GAMESS program [46]: electrostatic energy (E^es^), exchange energy (E^ex^), repulsive energy (E^rep^), polarization energy (E^pol^), and dispersion energy (E^disp^).

## 3. Results

The molecular electrostatic potential (MEP) of each of the monomers is illustrated in Figure 2. Of greatest importance is the positively charged σ-hole along the extension of the N-T bond axis of both S-CF_3_ and S-SiF_3_. The magnitude of the maximum of the MEP on the 0.001 au isodensity surface is equal to 13.8 and 40.8 kcal/mol, respectively, clearly much larger for the latter than for the C analog. The minimum of the MEP on each Lewis base occurs near its O atom and is equal to −40.8 kcal/mol for PO. Placing an electron-withdrawing group like CN or NO_2_ on the ring pulls density away from the O, thereby reducing the magnitude of V_min_. The reverse effect occurs for electron-donating substituents, which intensify this minimum. As may be seen by the V_min_ values listed in the first column of Table 1, this quantity ranges from −27.6 kcal/mol for PO-NO_2_ to −55.2 kcal/mol for PO-Li.

The pairing of either S-CF_3_ or S-SiF_3_ with each of the various bases leads to geometries exemplified by Figure 1 wherein the O atom of the base approaches the C or Si atom to form a tetrel bond. The salient features of the structures of these complexes reported in the remainder of Table 1 display some overarching trends. In the first place, the Si∙∙∙O TBs are considerably shorter than the C∙∙∙O bonds. The R_O_ distances from T to O in the former are all less than 2.0 Å, while they exceed 3.0 Å in the latter. Concomitant with the shorter Si∙∙∙O TBs is a much greater elongation of the internal R(N∙∙∙T) distance R_N_, which is roughly 0.1 Å, as compared to those for C∙∙∙O that are an order of magnitude smaller. Indeed, the R_O_/R_N_ ratio is nearly unity for the Si∙∙∙O interactions, signifying that the SiF_3_ group lies roughly midway between the N and O atoms. Along with what may be thought of as a half transfer, there is a flattening of the SiF_3_ group, in that the α angle in Figure 1 lowers from tetrahedral for the weak C∙∙∙O interactions to nearly 90° for Si∙∙∙O.

As a finer point, it may be noted that there is a relationship between the strength of the nucleophile as measured by its V_min_ and the geometric quantities. As this minimum intensifies, there is a general tendency for the TB to shorten, as is evident in the R_O_ quantities in Table 1. This contraction of TB induces an enhanced elongation of the internal R-N, ΔR_N_, as well as more flattening of the TF_3_ group as measured by α.

The interaction energy of each dimer E_int_ compares the energy of the complex with the sum of the individual monomers, held within their internal geometries they attain within the dimer. The binding energy E_b_ takes the fully optimized separated monomers as its reference point, so corresponds to the dimerization reaction energy. These two quantities differ by the energy DE required to deform each monomer into its geometry within the dimer. 

All of these quantities are contained in Table 2 for each of the complexes and are reflective of the geometrical trends in Table 1. The interaction and binding energies of the C∙∙∙O tetrel bonds in the upper half of Table 2 are only about 2–3 kcal/mol, consistent with the lengths of these weak intermolecular bonds that are greater than 3.0 Å. The highly compact Si∙∙∙O complexes contain much stronger bonds. Interaction energies range from 47 to 69 kcal/mol. Due to the distortion of the monomers incurred by the dimerization, with deformation energies between 30 and 40 kcal/mol, the binding energies are reduced to the 15–30 kcal/mol range, but still fall into the category of strong interactions. There is a strong correlation between the interaction energies and the Si∙∙∙O separations, as exhibited in Appendix A with a correlation coefficient of 0.99. Although imperfect, there is also a correlation observed between the interaction energetics and the minimum in the Lewis base’s MEP. As illustrated in Appendix A, this relationship is nearly linear with a correlation coefficient of 0.97 for the Si series.

One tool to better understand the nature and strength of these bonds arises in the context of AIM analysis of the electron density topology. The data reported in Table 3 refer to the bond critical points that lie roughly midway between the T atom and the O and N centers with which it interacts. In the case of the very weak tetrel bonds in the CF_3_ cases, the AIM procedure fails to locate a bond path between C and O, finding instead three paths from O to F, as pictured in Appendix A. This topology is reminiscent of other tetrel bonds involving –CF_3_ [4]. On the other hand, it is important to note that NBO analysis affirms the presence of a TB. In the S-CF_3_∙∙∙PO system, a charge transfer from the O lone pair of PO to the σ*(NC) antibonding orbital of S-CF_3_ is signaled by NBO via a second-order perturbation energy E^2^ of 0.35 kcal/mol, larger than any of the values corresponding to transfers to C-F antibonding orbitals. The situation is quite different for the SiF_3_ complexes in the lower half of Table 3. In the first place, there is a clear bond path connecting Si with O (see Appendix A), and its critical point density is between 0.07 and 0.08 au, an order of magnitude larger than in CF_3_. 

It is the sum of these three AIM quantities that are reported in the upper half of Table 3. Even with three separate terms contributing, the total of these three densities remains well below 0.01 au and the sum of the three energy densities is likewise quite small. The contrast with the far larger quantities on the right side of Table 3 is striking, with all indicating a strong covalent bond. The densities of the N-T bond in the SiF_3_ complexes drop significantly compared to CF_3_, down to around 0.10 au, a bit larger than the Si∙∙∙O quantities. The magnitudes of the three AIM properties contained in Table 3 are consistent with a strong noncovalent bond, with a certain degree of covalency [47]. In terms of patterns, the AIM quantities for the Si∙∙∙O bonds tend to rise as the substituent on the PO becomes more electron-donating, consistent with the geometric and energetic trends mentioned above.

An important component of the TB and related covalent bonds is the transfer of a certain amount of charge from the electron donor unit to the acceptor. The first column of Table 4 shows that the amount of this transfer (CT) is very small in the S-CF_3_ complexes, 0.001 e or less. But the transfer is greatly magnified in the SiF_3_ complexes approaching and surpassing 0.2 e. As in the case of the previous parameters, CT also rises as the PO substituent becomes more electron-donating toward the bottom of Table 4. This amount is greatly amplified in the S-SiF_3_ system, all close to 0.2 e. For each series of complex, whether S-CF_3_ or S-SiF_3_, CT follows a similar pattern with the interaction energy since both terms have a good linear relationship (Appendix A). In fact, Appendix A documents a close linear relationship between CT and E_int_, with a correlation coefficient of 0.99 for the Si systems.

The strong covalent elements in the bonds within the S-SiF_3_∙∙∙PO-X dimers precludes the usefulness of the NBO to extract the charge transfer between the N lone pair of PO-X and the σ*(N-Si) antibonding orbital [27] since the NBO views these complexes as a single molecular entity. Another useful tool that can be applied here derives from the Natural Orbitals for Chemical Valence (NOCV) approach. The isosurface of the NOCV pair density maps of S-CF_3_∙∙∙PO and S-SiF_3_∙∙∙PO is shown in Figure 3 as illustrative cases. The shapes of these density shift orbitals are similar but there is obviously a larger magnitude for the latter case. This idea of a larger magnitude carries over to the energies of the corresponding NOCV orbital energies listed in the last column of Table 4. In fact, an excellent linear relationship is found between the NOCV orbital energy and the full interaction energy, as evident by the correlation coefficient of 0.99 in Appendix A.

As another window into the nature of the T∙∙∙O TB, the interaction energy of each complex is decomposed into five terms, including electrostatic (E^es^), exchange (E^ex^), repulsion (E^rep^), polarization (E^pol^), and dispersion (E^disp^). As is clear from Table 5, the largest contributor to the weak C∙∙∙O TBs is the dispersion energy, which amounts to some 9 kcal/mol. The electrostatic and exchange terms are considerably smaller, and the polarization energy is only 1 kcal/mol. The magnitudes of these terms are much larger in the SiF_3_ systems. The dispersion energy climbs to nearly 30 kcal/mol but is superseded by the other contributors. The polarization energy rises to the 63–77 kcal/mol range, and the exchange energy is roughly 60 kcal/mol. The largest contribution originates in the electrostatic contribution, which amounts to some 75–100 kcal/mol. The near-dominance by E^es^ is consistent with the close correlation between the interaction energy and the minimum of the base’s MEP. But each of the various attractive terms is in good coincidence with the full interaction energy, as shown by the linear relationships in Appendix A, with correlation coefficients of E^es^, E^pol^, and E^disp^ equal to 0.98, 0.99, and 0.97, respectively.

## 4. Discussion

The forgoing data have documented that the formation of each complex elongates the internal R(N-T) distance within the S-TF_3_ subunit. While this stretch is only about 0.01 Å for T = C, it approaches 0.10 Å for the S-SiF_3_ systems. Given the roughly half transfer of the SiF_3_ groups in the latter complexes, one can take an alternate perspective and consider the stretch of the Si-O bond, from an imagined endpoint involving full transfer to the PO-X. The monomer resulting from this transfer depends upon whether the SiF_3_ group is electrically neutral or bears a positive charge, corresponding roughly to either a H atom or proton transfer within a H-bonded complex. Taking the unsubstituted PO unit as an example, the O-Si distance in the neutral SiF_3_-PO unit is optimized to 1.661 Å, while this same bond length is 1.703 Å in the cation. Since the Si∙∙∙O distance within the S-SiF_3_∙∙∙PO complex is shown to be 1.874 Å in Table 1, the latter distance can be considered a stretch of 0.213 Å relative to the neutral, and 0.171 Å with respect to the cationic monomer. Both of these stretches are somewhat larger than the elongation of the N-Si bond of 0.092 Å relative to the S-SiF_3_ monomer. So, from this geometrical perspective, the degree of SiF_3_ transfer can be thought of as roughly 30% for the former neutral case and 35% for the cation transfer.

In biological and organic molecules, alkyl groups are commonly bonded to electronegative atoms. Hence, the interactions involving these alkyl groups, although individually weak, may be commonplace and together play an important cumulative role in the overall structure and function [13]. It is known that the transition state of S_N_2 reactions usually involves a quasi-pentacoordinate C interacting with two nucleophilic groups, so carbon bonding may facilitate this class of reaction [4]. This idea has motivated some of the attention that has lately been paid to the carbon bonding of methyl groups [13,19,48,49]. A consensus has been reached that C-bonding of any strength requires the presence of at least one electron-withdrawing substituent on the Lewis acid. The replacement of the three H atoms of CH_3_ by F does not guarantee a deeper σ-hole nor a stronger tetrel bond [27], which may be due to the repulsion between the three F atoms and the electron donor atom. The choice of saccharin (1,2-benzisothiazol-3(2H)-one-1,1-dioxide) as the unit to which the TF_3_ group is attached here is guided in part by the presence of two electron-withdrawing groups (carbonyl and sulfonyl) attached to the N atom, which is itself electronegative [27]. These factors contribute to the deep σ-hole on C for which V_max_ is 13.8 kcal/mol, and 40.8 if C is replaced by Si. In fact, the calculations delineated here verify that a tetrel bond is formed between the CF_3_ group and the PO derivatives, with interaction energies between 2 and 3 kcal/mol.

Of course, bonds of this type can be magnified by charge assistance. That is, if the Lewis acid containing –CF_3_ is positively charged [27] or the electron donor is an anion [50,51], the tetrel bond formed would be strengthened, with past results leading to values in the 7–8 kcal/mol range. A less extreme bond strengthening occurs if the base is enhanced by a strong electron donor as in the case of the metal atoms of NCLi and NCNa [4,27], in which case the C∙∙∙N TB involving a CF_3_ group rises up to about 4 kcal/mol. The choice of pyridine-1-oxide (PO) and its derivatives as a Lewis base here is made in part due to the strong potential of N-oxides as an electron donor. For example, past work found an interaction energy of 52 kcal/mol [52] upon pairing H_3_NO with PhSiF_3_ and PhSiH_3_.

Whereas the mutation of the –CH_3_ electron acceptor to –CF_3_ does not necessarily strengthen the TB [27], the analogous fluorination of SiH_3_ does so, as, for example, the rise in the bond strength by 33 kcal/mol when H_3_NO acts as electron donor [52]. When a series of anions F^−^, Cl^−^, Br^−^, CN^−^, NC^−^, N_3_^−^, NCS^−^, and SCN^−^ bind with NCCF_3_ and NCSiF_3_ through a tetrel bond, the increase in interaction energy lies in the range of 18–75 kcal/mol when the C of NCCF_3_ is changed to Si in NCSiF_3_ [51]. With neutral PO-X as an electron donor, the TB strengthening on going from C to Si ranges from 44 to 67 kcal/mol.

The presence of a σ-hole on the tetrel atom can offer a reliable prediction for the formation of a tetrel bond, but its magnitude does not always correspond accurately with the bond strength. For example, in tetrel-bonded complexes of formamidine with TH_3_F (T = C, Si, Ge, and Sn), the interaction energy increases in the order C < Ge < Si < Sn, inconsistent with the depth of the σ-hole on the T atom [53]. Similarly, while the negative MEP on the electron donor atom is predictive of its ability to form a tetrel bond, there are exceptions. The top half of Table 2 provides a case in point where the interaction energies do not follow the same pattern as does V_min_ for the S-CF_3_∙∙∙PO-X complexes. This inconsistency fits with prior reports of the tetrel-bonded complexes of TF_4_ (T = Si, Ge, and Sn) with pyrazine and HCN [54], as well as in the halogen-bonded complexes of CF_3_Cl with methylated amines [55]. In the case considered here, the inconsistency may be partly attributed to the stabilizing interactions other than the TB, which are marked by the AIM bond paths to the F atoms.

As described above, an electron-withdrawing substituent placed on the PO-X Lewis base attenuates its V_max_, while the opposite occurs for an electron-donating X. This trend bears a direct consequence for the interaction energy. Figure 4 plots E_int_ for the S-SiF_3_∙∙∙POX systems against the aromatic substituent constant (δ) in a pair of Hammett plots, which document the strong relationship between these two parameters. It is hoped that a similar correlation can be helpful in estimating the strength of similar bonds in other types of noncovalent interactions, tetrel bonds or otherwise [56]. It is worth noting that an analogous relationship has been reported for the substituents in the Lewis acid molecule [57]. 

Because of its strong electron-releasing characteristic, the Li substituent leads to the strongest tetrel bonding enhancement when placed on the Lewis base. Extrapolating this idea, an alkali metal substituent may make the methyl C atom nucleophilic [9]. It stands to reason that two alkali metal atoms would make this enhancing effect even more prominent. To substantiate this idea, the interaction energy between CH_3_F and C_2_H_2_ is only 1 kcal/mol, but two Li/Na substituents elevate this quantity by a factor of 5 to 5–6 kcal/mol [58]. Of course, other metals also have a similar effect. The tetrel bond formed between NCH and SiF_4_ is weak with an interaction energy of 3.45 kcal/mol, but is increased nearly fourfold by the change from NCH to NCAg [59]. 

Our own group has been attracted to the question as to whether and under what circumstances a TR_3_ group may be transferred across a tetrel bond, given its relevance to the S_N_2 reaction and its significance in biological systems. It was learned that N-heterocyclic carbene (C_3_N_2_H_4_, NHC) is a good electron donor in a tetrel bond (the interaction energy amounts to 26 kcal/mol with SiH_3_F as the Lewis acid) [60], but NHC forms a weak tetrel bond (3.9 kcal/mol) with PhSiH_3_ [34]. If Be^2+^ is added above the ring plane, resulting in a cation–π interaction, the interaction energy rises to 100 kcal/mol, and this strong tetrel bond results in the displacement of the –SiH_3_ group from the C atom to N [34]. A similar cooperative effect occurs with a triel bond that can also aid in this transfer [27]. In the binary complexes between TF_3_OH (T = Si and Ge) and Lewis base NCM (M = Li and Na), a half-transfer of –TF_3_ occurs when a third BeCl_2_ molecule approaches the O atom of TF_3_OH and a fourth NCM molecule is introduced simultaneously [35]. The aforementioned transfers all rely on the cooperativity of the tetrel bond with other noncovalent interactions. The results described above deviate from this pattern in that the SiF_3_ transfer occurs without need of the cooperativity of a third unit. This phenomenon may be due in part to the weakness of the internal N-Si bond in comparison to Si-O, coupled with the presence of the electron-withdrawing groups on the Lewis acid. This promotion is coupled with the strong nucleophilic characteristic of the N-oxide of the base used here.

## 5. Conclusions 

A tetrel bond is formed between the O atom of pyridine-1-oxide (PO) and the tetrel atom T, either C or Si, on 1,2-benzisothiazol-3-one-2-TF_3_-1,1-dioxide. Electron-withdrawing substituents placed on the former tend to weaken this TB as they make the electron pair of the donor O less accessible; the reverse occurs for electron-donating groups. The Si∙∙∙O TBs are far stronger than their C∙∙∙O analogs, reaching up toward 70 kcal/mol. Formation of the C∙∙∙O TB amounts to less than 3 kcal/mol and, occasionally, a small stretch of the internal N-C covalent bond. The Si∙∙∙O interaction is strong enough that the SiF_3_ unit translates to a position roughly halfway between the N and O atoms. 

## Figures and Tables

**Figure 1 ijms-24-11884-f001:**
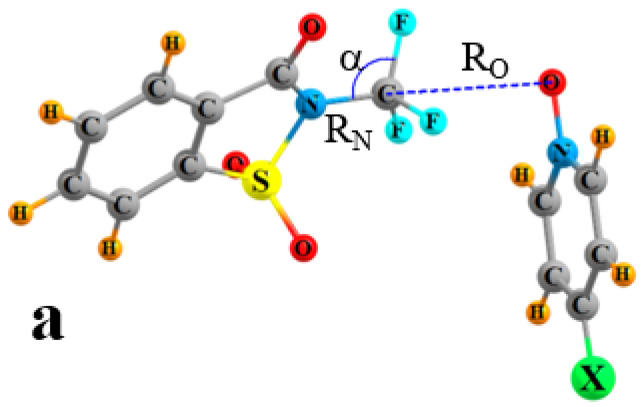
Geometries of complexes (**a**) S-CF_3_ and (**b**) S-SiF_3_ with PO-X bases, indicating definitions of geometrical parameters.

**Figure 2 ijms-24-11884-f002:**
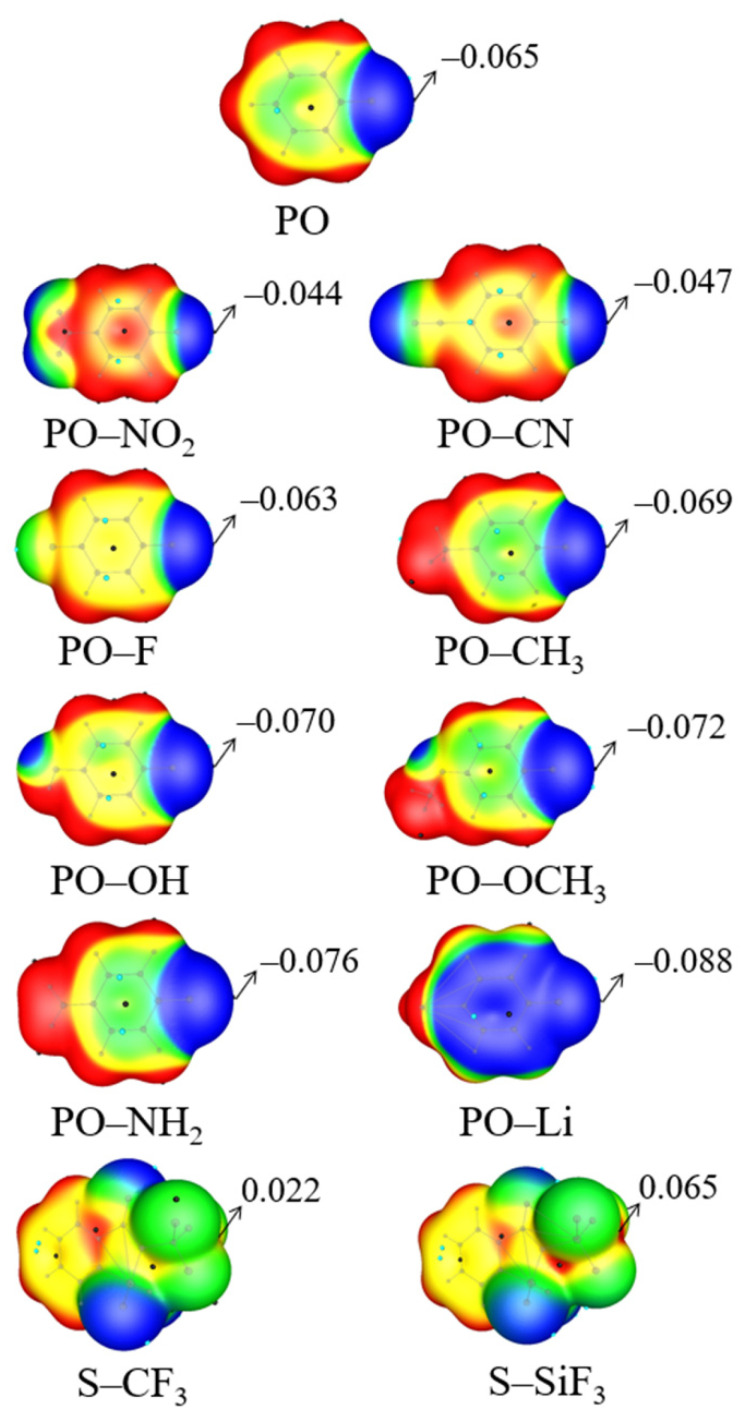
MEP maps of PO–X and S–TF_3_ monomers. Color ranges in a.u: red, greater than 0.02; yellow, between 0.02 and 0; green, between −0.02 and 0; blue, less than −0.02.

**Figure 3 ijms-24-11884-f003:**
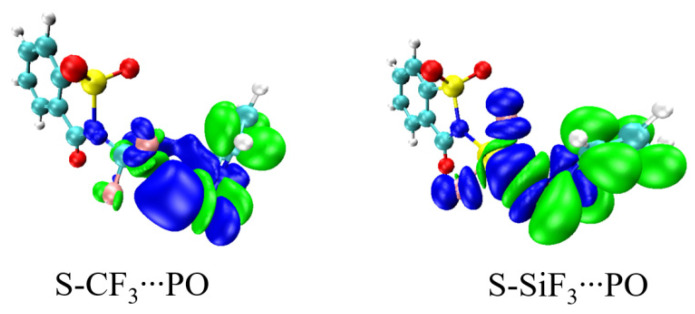
Isosurface of NOCV pair density map of S-CF_3_∙∙∙PO and S-SiF_3_∙∙∙PO.

**Figure 4 ijms-24-11884-f004:**
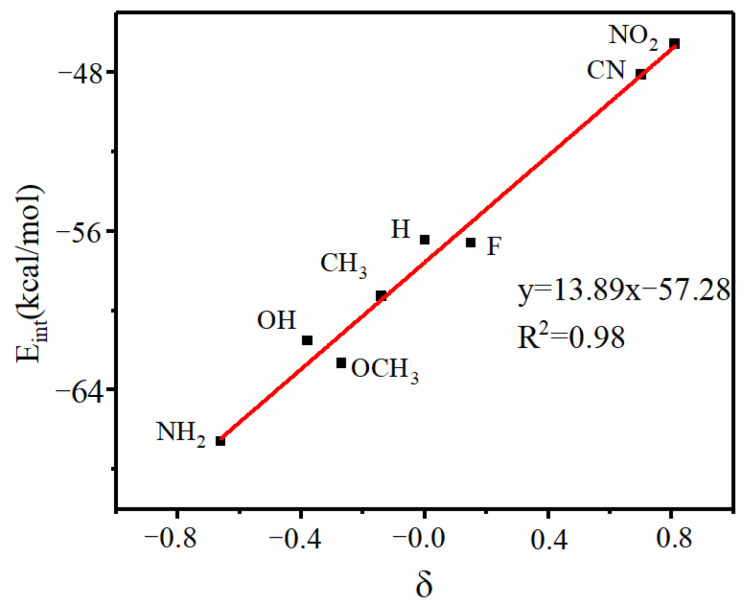
Hammett plots of S-SiF_3_∙∙∙POX complexes.

**Table 1 ijms-24-11884-t001:** Minima in MEP of Lewis bases (V_min_, kcal/mol), T∙∙∙O distance (R_O_, Å), N-T bond length (R_N_, Å), and its change (ΔR_N_) relative to the monomer, average of three N-T-F angles (α, deg) in the complexes.

	V_min_	R_O_	R_N_	ΔR_N_	R_O_/R_N_	γ_O_ ^a^	γ_N_ ^a^	α
S-CF_3_∙∙∙PO-NO_2_	−27.6	3.123	1.431	0.005	2.183	0.970	0.441	110.2
S-CF_3_∙∙∙PO	−40.8	3.151	1.436	0.010	2.193	0.979	0.442	109.9
S-CF_3_∙∙∙PO-CH_3_	−43.3	3.119	1.437	0.011	2.169	0.969	0.442	109.8
S-CF_3_∙∙∙PO-OH	−43.9	3.115	1.437	0.011	2.169	0.967	0.442	109.8
S-CF_3_∙∙∙PO-OCH_3_	−45.2	3.144	1.437	0.011	2.188	0.976	0.442	109.8
S-CF_3_∙∙∙PO-NH_2_	−47.7	3.085	1.438	0.012	2.146	0.958	0.442	109.7
S-CF_3_∙∙∙PO-Li	−55.2	3.097	1.440	0.014	2.151	0.962	0.443	109.6
S-SiF_3_∙∙∙PO-NO_2_	−27.6	1.902	1.819	0.080	1.045	0.526	0.499	94.3
S-SiF_3_∙∙∙PO-CN	−29.5	1.896	1.821	0.082	1.042	0.524	0.499	94.2
S-SiF_3_∙∙∙PO-F	−39.5	1.874	1.829	0.090	1.025	0.518	0.501	93.5
S-SiF_3_∙∙∙PO	−40.8	1.874	1.831	0.092	1.024	0.518	0.502	93.3
S-SiF_3_∙∙∙PO-CH_3_	−43.3	1.866	1.834	0.095	1.017	0.515	0.502	93.1
S-SiF_3_∙∙∙PO-OH	−43.9	1.860	1.834	0.095	1.014	0.514	0.502	93.0
S-SiF_3_∙∙∙PO-OCH_3_	−45.2	1.856	1.836	0.097	1.011	0.513	0.503	92.9
S-SiF_3_∙∙∙PO-NH_2_	−47.7	1.848	1.841	0.102	1.004	0.510	0.504	91.2
S-SiF_3_∙∙∙PO-Li	−55.2	1.842	1.849	0.110	0.996	0.509	0.507	92.2

^a^ γ is the ratio of R_O_ or R_N_ to the sum of the van der Waals radii of the two atoms.

**Table 2 ijms-24-11884-t002:** Interaction energy (E_int_), binding energy (E_b_), and deformation energy (DE) in the complexes, all in kcal/mol.

	E_int_	E_b_	DE
S-CF_3_∙∙∙PO-NO_2_	−2.69	−2.55	0.14
S-CF_3_∙∙∙PO	−2.33	−2.12	0.21
S-CF_3_∙∙∙PO-CH_3_	−2.36	−2.13	0.23
S-CF_3_∙∙∙PO-OH	−2.63	−2.35	0.28
S-CF_3_∙∙∙PO-OCH_3_	−2.62	−2.36	0.26
S-CF_3_∙∙∙PO-NH_2_	−2.37	−2.11	0.26
S-CF_3_∙∙∙PO-Li	−2.01	−1.62	0.39
S-SiF_3_∙∙∙PO-NO_2_	−46.59	−15.12	31.47
S-SiF_3_∙∙∙PO-CN	−48.12	−16.29	31.83
S-SiF_3_∙∙∙PO-F	−56.59	−22.39	34.20
S-SiF_3_∙∙∙PO	−56.45	−21.82	34.63
S-SiF_3_∙∙∙PO-CH_3_	−59.29	−23.78	35.51
S-SiF_3_∙∙∙PO-OH	−61.52	−25.64	35.88
S-SiF_3_∙∙∙PO-OCH_3_	−62.65	−26.29	36.36
S-SiF_3_∙∙∙PO-NH_2_	−66.58	−28.50	38.08
S-SiF_3_∙∙∙PO-Li	−69.15	−29.76	39.39

**Table 3 ijms-24-11884-t003:** Electron densities (ρ), Laplacians (∇^2^ρ), and energy densities (H) at the T···O and N-T BCPs in the complexes, all in a.u.

	T∙∙∙O ^a^			N-T		
	ρ	∇^2^ρ	H	ρ	∇^2^ρ	H
S-CF_3_∙∙∙PO-NO_2_	0.0067	0.0372	0.0018	0.2926	−0.8536	−0.3242
S-CF_3_∙∙∙PO	0.0072	0.0390	0.0016	0.2894	−0.8362	−0.3191
S-CF_3_∙∙∙PO-CH_3_	0.0074	0.0403	0.0016	0.2885	−0.8316	−0.3177
S-CF_3_∙∙∙PO-OH	0.0066	0.0361	0.0018	0.2885	−0.8312	−0.3176
S-CF_3_∙∙∙PO-OCH_3_	0.0066	0.0357	0.0017	0.2882	−0.8294	−0.3169
S-CF_3_∙∙∙PO-NH_2_	0.0078	0.0423	0.0017	0.2876	−0.8271	−0.3166
S-CF_3_∙∙∙PO-Li	0.0071	0.0388	0.0018	0.2853	−0.8170	−0.3142
S-SiF_3_∙∙∙PO-NO_2_	0.0707	0.3233	−0.0162	0.1004	0.4831	−0.0279
S-SiF_3_∙∙∙PO-CN	0.0716	0.3317	−0.0162	0.1000	0.4796	−0.0278
S-SiF_3_∙∙∙PO-F	0.0758	0.3686	−0.0163	0.0982	0.4656	−0.0272
S-SiF_3_∙∙∙PO	0.0756	0.3680	−0.0161	0.0976	0.4612	−0.0269
S-SiF_3_∙∙∙PO-OH	0.0783	0.3917	−0.0163	0.0969	0.4551	−0.0267
S-SiF_3_∙∙∙PO-NH_2_	0.0808	0.4139	−0.0164	0.0955	0.4444	−0.0263
S-SiF_3_∙∙∙PO-OCH_3_	0.0790	0.3990	−0.0162	0.0965	0.4524	−0.0266
S-SiF_3_∙∙∙PO-NH_2_	0.0808	0.4139	−0.0164	0.0955	0.4444	−0.0263
S-SiF_3_∙∙∙PO-Li	0.0815	0.4247	−0.0161	0.0937	0.4308	−0.0257

^a^ Topological parameters at the intermolecular BCP in the S-CF_3_ complexes were obtained as the sum of three F∙∙∙O BCPs.

**Table 4 ijms-24-11884-t004:** Charge transfer (CT, e) and NOCV orbital energies (E, kcal/mol) in the complexes.

	CT ^a^	E
S-CF_3_∙∙∙PO-NO_2_	0.0013	−0.31
S-CF_3_∙∙∙PO	0.0009	−0.36
S-CF_3_∙∙∙PO-CH_3_	0.0004	−0.39
S-CF_3_∙∙∙PO-OH	0.0001	−0.39
S-CF_3_∙∙∙PO-OCH_3_	0.0006	−0.39
S-CF_3_∙∙∙PO-NH_2_	0.0005	−0.44
S-CF_3_∙∙∙PO-Li	0.0002	−0.43
S-SiF_3_∙∙∙PO-NO_2_	0.1755	−54.50
S-SiF_3_∙∙∙PO-CN	0.1780	−55.26
S-SiF_3_∙∙∙PO-F	0.1900	−58.50
S-SiF_3_∙∙∙PO	0.1907	−58.67
S-SiF_3_∙∙∙PO-CH_3_	0.1942	−60.00
S-SiF_3_∙∙∙PO-OH	0.1961	−60.58
S-SiF_3_∙∙∙PO-OCH_3_	0.1977	−61.20
S-SiF_3_∙∙∙PO-NH_2_	0.2024	−62.59
S-SiF_3_∙∙∙PO-Li	0.2059	−64.08

^a^ Sum of natural atomic charges of PO-X subunit.

**Table 5 ijms-24-11884-t005:** Electrostatic (E^es^), exchange (E^ex^), repulsion (E^rep^), polarization (E^pol^), and dispersion energies (E^disp^), as well as the total interaction energy (ΔE^total^) in the complexes. All in kcal/mol.

	E^es^	E^ex^	E^rep^	E^pol^	E^disp^	ΔE^total^
S-CF_3_∙∙∙PO-NO_2_	−3.61	−2.77	12.79	−0.74	−8.45	−2.78
S-CF_3_∙∙∙PO	−3.37	−3.12	13.68	−0.88	−8.69	−2.38
S-CF_3_∙∙∙PO-CH_3_	−3.50	−3.30	14.17	−1.03	−8.76	−2.42
S-CF_3_∙∙∙PO-OH	−3.70	−3.23	13.92	−1.04	−8.68	−2.73
S-CF_3_∙∙∙PO-OCH_3_	−3.64	−3.26	14.11	−1.05	−8.86	−2.70
S-CF_3_∙∙∙PO-NH_2_	−3.80	−3.69	15.17	−1.14	−8.95	−2.42
S-CF_3_∙∙∙PO-Li	−2.97	−3.28	13.73	−1.24	−8.31	−2.06
S-SiF_3_∙∙∙PO-NO_2_	−74.86	−54.07	169.62	−63.03	−26.97	−49.31
S-SiF_3_∙∙∙PO-CN	−76.91	−54.94	172.17	−64.04	−27.17	−50.89
S-SiF_3_∙∙∙PO-F	−88.66	−59.35	185.02	−68.96	−27.67	−59.62
S-SiF_3_∙∙∙PO	−87.17	−58.59	182.73	−68.78	−27.68	−59.51
S-SiF_3_∙∙∙PO-CH_3_	−90.66	−59.97	186.83	−70.93	−27.75	−62.48
S-SiF_3_∙∙∙PO-OH	−94.57	−61.55	191.51	−72.18	−27.92	−64.72
S-SiF_3_∙∙∙PO-OCH_3_	−95.80	−62.06	193.06	−73.16	−27.90	−65.85
S-SiF_3_∙∙∙PO-NH_2_	−100.42	−63.69	197.82	−75.37	−28.25	−69.91
S-SiF_3_∙∙∙PO-Li	−100.95	−63.45	196.96	−77.16	−28.09	−72.69

## Data Availability

The data that support the findings of this study are available within the article and its Appendix A.

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
