# Peer review of "C∙∙∙O and Si∙∙∙O Tetrel Bonds: Substituent Effects and Transfer of the SiF3 Group"

_ijms, 2023, doi:10.3390/ijms241511884_

Round 1

Reviewer 1 Report

The Authors have presented a very interesting research in a concise way. The methods are described properly, pointing out all the necessary technical details. All graphs and tables are easy to read. I do not see a huge space for any improvement, as the manuscript is already prepared properly. I may have only almost neglectable comments about typing errors: in line 307 you mention ''Hammet plots'' and below the graph in line 313 it is ''Hammett plots''. And in the title there should be a space between ''bonds.'' and ''Substituent''. 

Author Response

Response: The right is Hammett plots, which has been corrected. The space between ''bonds.'' and ''Substituent'' has been inserted. Thanks for the high comment for our paper.

Reviewer 2 Report

The authors performed the research using topological analysis of electron density (Bader's theory). The authors used the augmented Dunning basis sets. A detailed analysis of the calculated electron density parameters is performed. The results obtained are of interest, so it is recommended to publish the article as is.

Author Response

Response: Thanks for this positive comment.

Reviewer 3 Report

This beautifully written paper by Scheiner et al. is a fine computational study on tetrel bonding.  The authors performed a methodologically thorough investigation and comparison of Si...O and C...O tetrel bonds with pyridine-1-oxide derivatives. Even though the results regarding the substituent effects are expected (electron-withdrawing substituent weakens the tetrel bonding and electron-donating substituent strengthens it), it was actually very interesting to see the transfer of SiF3 group without a help of another interaction, which had huge influence on binding and interaction energies. 

This paper needs to go through a revision in order to be accepted for publication. This includes the following points.

1. Why did the authors study tetrel bonds with PO-F and PO-CN only for S-SiF3 system, and not for S-CF3 system as well? The C..O tetrel bonds with these PO derivatives need to be added in order to complete this research. 

2. MEP maps of monomers should be transferred to the main text.

3. It would be more sound if the paper had separate Conclusion section. 

Author Response

Comment 1: Why did the authors study tetrel bonds with PO-F and PO-CN only for S-SiF3 system, and not for S-CF3 system as well? The C...O tetrel bonds with these PO derivatives need to be added in order to complete this research.

Response: Actually, both S-CF3 systems of PO-F and PO-CN have also been optimized, but the corresponding tetrel-bonded structures are not obtained since the tetrel bond is very weak for these systems, thus their geometries and properties are not investigated.

Comment 2: MEP maps of monomers should be transferred to the main text.

Response: This figure has been moved to the main text as Figure 2.

Comment 3: It would be more sound if the paper had separate Conclusion section.

Response: Conclusion has been separated and added.

Round 2

Reviewer 3 Report

This is a very good paper and in my opinion it should be accepted in the present form.